# Pre-trained Language Models Improve the Few-shot Prompt Ability of Decision Transformer

**Yu Yang**                                                                *yu.yang@duke.edu*
*Department of Electrical and Computer Engineering*
*Duke University*

**Pan Xu**                                                                 *pan.xu@duke.edu*
*Department of Biostatistics and Bioinformatics*
*Department of Computer Science*
*Department of Electrical and Computer Engineering*
*Duke University*

**Reviewed on OpenReview:** *https://openreview.net/forum?id=k520i3XEMK*

## Abstract

Decision Transformer (DT) has emerged as a promising class of algorithms in offline reinforcement learning (RL) tasks, leveraging pre-collected datasets and Transformer's capability to model long sequences. Recent works have demonstrated that using parts of trajectories from training tasks as prompts in DT enhances its performance on unseen tasks, giving rise to Prompt-DT methods. However, collecting data from specific environments can be both costly and unsafe in many scenarios, leading to suboptimal performance and limited few-shot prompt abilities due to the data-hungry nature of Transformer-based models. Additionally, the limited datasets used in pre-training make it challenging for Prompt-DT type of methods to distinguish between various RL tasks through prompts alone. To address these challenges, we introduce the Language model-initialized Prompt Decision Transformer (LPDT) framework, which leverages pretrained language models providing rich prior knowledge for RL tasks and fine-tunes the sequence model using Low-rank Adaptation (LoRA) for meta-RL problems. We further incorporate prompt regularization to effectively differentiate between tasks based on prompt feature representations. Comprehensive empirical studies demonstrate that initializing with a pre-trained language model provides the prior knowledge and achieves a similar performance with Prompt-DT under only 10% data in some MuJoCo control tasks. We also provide a thorough ablation study to validate the effectiveness of each component, including sequence modeling, language models, prompt regularizations, and prompt strategies.

## 1 Introduction

In many sequential decision-making applications such as robotic manipulation and autonomous driving (Sinha et al., 2022; Kumar et al., 2022), it can be expensive or even unsafe for agents to learn through trial-and-error in the environment. Offline reinforcement learning (RL) methods (Levine et al., 2020) have emerged as a powerful paradigm for optimizing agent policies without extensive online interaction. These methods learn an optimal policy by leveraging pre-collected datasets obtained from a set of behavior policies. Among them, Decision Transformer (DT) (Chen et al., 2021) and its successors (Wu et al., 2024; Zhuang et al., 2024) have become popular due to their scalability and training stability. DT models a return-conditioned policy using the powerful Transformer architecture, solving RL as a sequence prediction problem in a supervised learning manner. It models the states, actions, and return-to-go from trajectories as the tokens of an input sequence, and generates actions conditioned on the return-to-go. Compared with dynamic programming-based offline RL methods (Kumar et al., 2019; Fujimoto et al., 2019; Kumar et al., 2020) that heavily rely on

the Markov Decision Process (MDP) assumption, DT can utilize entire trajectory histories to predict the next action, making it more applicable in partially observable environments where all past information is crucial for decision-making (Kaelbling et al., 1998; Ni et al., 2024). Furthermore, the supervised learning nature of DTs enhances the stability and scalability in the training process compared to dynamic programming algorithms based on Bellman equations (Chen et al., 2021; Janner et al., 2021; Zheng et al., 2022; Wu et al., 2024; Zhuang et al., 2024).

Transformers have demonstrated remarkable few-shot generalization capabilities, most notably in large language models (LLMs) (Brown et al., 2020; Achiam et al., 2023). In the context of LLMs, prompt-based learning, where task-relevant information is provided as a textual prefix, has been proven highly effective for adapting to new tasks without fine-tuning (Brown et al., 2020; Li & Liang, 2021). Inspired by this, recent works (Xu et al., 2022; Hu et al., 2023; Wang et al., 2024b) have developed variants of DT to leverage trajectory-based prompting learning to enhance its few-shot generalization. For instance, Prompt-DT (Xu et al., 2022) leverages segments of trajectories from offline datasets as prompts to encode task-specific information. Similar to how large language models (LLMs) conditioned on textual prompts to generate coherent responses, Prompt-DT conditions on trajectory-based prompts to generate actions. The model is trained on these prompt-trajectory pairs and is subsequently evaluated on unseen tasks using few-shot demonstrations as prompts. However, existing Prompt-DT methods (Xu et al., 2022; Hu et al., 2023; 2024; Wang et al., 2024b) inherit the data-hungry nature of Transformers (Brown et al., 2020; Achiam et al., 2023), requiring training on large trajectory datasets, while offline RL datasets are often too small to fully realize their few-shot prompting potential. In contrast, modern LLMs achieve strong generalization with limited task examples by leveraging massive unsupervised pre-training on text. Inspired by this, another line of works has investigated using pre-trained language models to initialize decision transformers and transfer their rich linguistic priors to RL domains (Li et al., 2022; Reid et al., 2022; Shi et al., 2024). While this initialization provides valuable prior knowledge and helps alleviate the need for large datasets, existing studies have primarily focused on single-task RL problems. The potential for enhancing few-shot prompt learning of DTs in multi-task settings using pre-trained language models remains unexplored.

To overcome these limitations, we propose a novel framework named *Language model-initialized Prompt Decision Transformer (LPDT)*. Our approach leverages a pre-trained language model for initialization to significantly improve the few-shot prompting capabilities of Decision Transformers. This incorporates pre-existing knowledge that can benefit downstream RL tasks. To efficiently combine this pre-trained knowledge with domain-specific knowledge from multi-task RL, we use Low-Rank Adaptation (LoRA) (Hu et al., 2021), a parameter-efficient fine-tuning method. We further introduce prompt regularization methods to help the fine-tuned model distinguish different RL tasks, thereby guiding action generation based on new, unseen task-specific prompt representations. A more detailed illustration of our model structure and training paradigm is provided in Figure 1. Beyond Decision Transformer, we also extend our framework to one of its most recent and advanced variants, Reinformer (Zhuang et al., 2024), demonstrating its flexibility. We conduct extensive experiments to assess the capability of our proposed framework in MuJoCo control environments (Fu et al., 2020) and Meta-World ML1 tasks (Yu et al., 2020). Our method outperforms baselines in terms of cumulative rewards on unseen tasks. Empirical studies show that our proposed framework achieves similar performance to Prompt-DT using only 10% of the dataset. We also provide a detailed ablation study to demonstrate the effectiveness of our LPDT framework. Specifically, we evaluate our proposed framework to validate each component. Additionally, we report results on various language model initializations and different sequence modeling strategies. Furthermore, we design experiments to demonstrate that the improved performance stems from better initialization rather than from the language model's inherent understanding of RL tasks.

- We propose LPDT, a novel framework that improves the few-shot prompt capabilities of Decision Transformers and other sequence modeling methods for offline RL. Our approach leverages a pre-trained language model for initialization and incorporates a novel prompt regularization method, demonstrating enhanced few-shot generalization in multi-task settings.
- We introduce a unique combination of Low-Rank Adaptation (LoRA) and prompt regularization methods to effectively combine pre-trained knowledge with domain-specific RL task knowledge. LoRA allows efficient fine-tuning by adapting a small subset of parameters, while our regularization methods enhance the model's ability to distinguish task information within prompts.

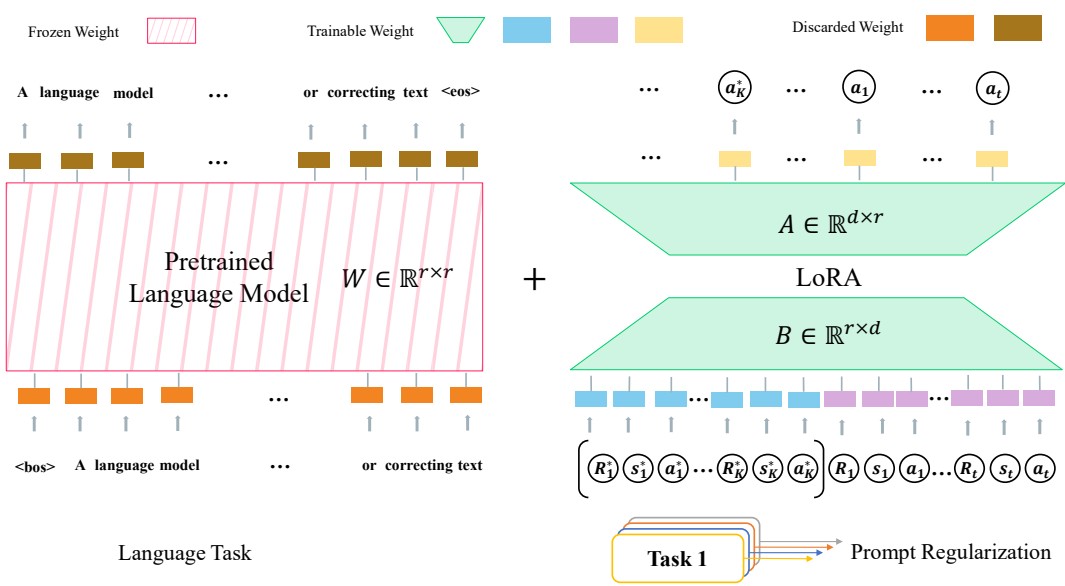

Figure 1: Overview of LPDT. We first initialize our algorithm using a pre-trained language model. The pre-trained language model is trained on a large text corpus using the causal language modeling objective, i.e., predicting the next token. Our method LPDT replaces the word embedding layers with linear layers, discarding the learned word features, to fully learn and capture the features of RL trajectory tokens. We fine-tune our model using parameter-efficient methods like Low-Rank Adaptation (LoRA). Specifically, we freeze the initial weights of the language model and update only the LoRA weights. The input to our approach consists of prompts accompanied by training trajectories from the same tasks. Unlike traditional models that predict word tokens, our method predicts action tokens in the RL trajectories. Additionally, we incorporate prompt regularization on the input prompts. This is achieved by introducing an additional loss on the prompt embeddings, which helps LPDT distinguish between different environments. More technical details of our method are presented in Section 3.

- We extend our proposed framework, LPDT, to one of the most recent and advanced variants of DT, i.e., Reinformer, to demonstrate its generalizability.
- Through extensive experiments on MuJoCo control and Meta-World ML1, we show that LPDT outperforms baselines on unseen tasks and achieves similar performance to Prompt-DT with only 10% of the data in Cheetah-vel and Ant-dir. We also validate each component of our framework through detailed ablation studies.

## 2 Preliminary

### 2.1 Offline Reinforcement Learning

Reinforcement learning is usually formulated as solving a Markov Decision Process (MDP) defined by a tuple $(\mathcal{S}, \mathcal{A}, \mathcal{T}, d_0, \mathcal{R}, \gamma)$, where $\mathcal{S}$ represents the set of states $s \in \mathcal{S}$, $\mathcal{A}$ represents the set of actions $a \in \mathcal{A}$, $\mathcal{T}$ is the transition distribution defined as $\mathcal{T}(s_{t+1}|s_t, a_t)$, $d_0$ is the distribution of initial states $s_0$, $\mathcal{R} : \mathcal{S} \times \mathcal{A} \to \mathbb{R}$ is the reward function, $r_t = \mathcal{R}(s_t, a_t)$ is the reward at timestep $t$, and $\gamma \in (0, 1)$ is the discount factor. The objective is to find a policy $\pi$ that maximizes the expected cumulative rewards $J(\pi)$:

$$J(\pi) = \mathbb{E}_{s_0 \sim d_0(\cdot), a_t \sim \pi(\cdot|s_t), s_{t+1} \sim \mathcal{T}(\cdot|s_t, a_t)} \left[ \sum_{t=0}^{\infty} \gamma^t \mathcal{R}(s_t, a_t) \right].$$

In offline RL, the agent has access to a dataset $\mathcal{D}$ containing trajectories collected by a behavior policy instead of access to the environment. The agent is expected to find the optimal policy using only the offline dataset $\mathcal{D}$, without interacting with the environment itself. Decision Transformer (Chen et al., 2021) leverages the Transformer structure (Vaswani et al., 2017) to predict the next action conditioned

on the past trajectory. To reduce the prediction error that could accumulate as the trajectory length increases, DT reformulates the trajectories $\{s_0, a_0, r_0, s_1, a_1, r_1, \ldots, s_T, a_T, r_T\}$ from the offline dataset $\mathcal{D}$ to $\{s_0, a_0, R_0, s_1, a_1, R_1, \ldots, s_T, a_T, R_T\}$ and uses the latter one in the loss function defined by the prediction error between the true actions and predicted actions, where $R_t = \sum_{i=t}^{T} r_i$ is the return-to-go at timestep $t$.

## 2.2 Prompt Decision Transformer

The goal of Prompt-DT is to enable a single offline RL agent to solve multiple distinct tasks without requiring explicit task labels. The key idea is to prepend a prompt $\tau_i^*$ to the standard DT input trajectory $\tau$. Both the prompt and the main trajectory are sampled from the same multi-task offline dataset $\mathcal{D}$. A prompt $\tau_i^*$ for a task $T_i$ is a short sequence of $K^*$ transitions denoted as $\tau_i^* = \left(R_{i,1}^*, s_{i,1}^*, a_{i,1}^*, \cdots, R_{i,K^*}^*, s_{i,K^*}^*, a_{i,K^*}^*\right)$. This prompt provides the model with crucial context about the task's dynamics and reward structure. Consequently, the input vector $\tau_i^{\text{input}}$ is the concatenation of the task prompt and the input trajectory:

$$\tau_i^{\text{input}} = [\tau_i^*, \tau_i] = \left(R_{i,1}^*, s_{i,1}^*, a_{i,1}^*, \cdots, R_{i,K^*}^*, s_{i,K^*}^*, a_{i,K^*}^*, R_{i,1}, s_{i,1}, a_{i,1}, \ldots, R_{i,K}, s_{i,K}, a_{i,K}\right). \quad (2.1)$$

In addition to this, we denote the trajectory truncated at timestep $t \in \{1, \ldots, K\}$ as $\tau_{i<t}$, and consequently the input vector at timestep $t$ as $\tau_{i,1<t}^{\text{input}}$. Then the learning objective of Prompt-DT can be formulated as the following maximum likelihood estimation: $\mathbb{E}_{\tau_i^{\text{input}} \sim T_i}\left[\sum_{t=1}^{K} -\log M_\theta(\hat{a}_{i,t} | \tau_i^*, \tau_{i,1<t-1}^{\text{input}}, R_{i,t}, s_{i,t})\right]$, where $M_\theta$ denotes Prompt-DT with the parameter $\theta$ which is usually a transformer. In practical implementation, we often use the mean squared error loss instead, which aims to predict the future action $\hat{a}_{i,t}$ given the history trajectory and current state by minimizing the loss function $\mathcal{L}_{\text{PDT}} = \mathbb{E}_{\tau_i^{\text{input}} \sim T_i}\left[\frac{1}{K}\sum_{t=1}^{K}(a_{i,t} - \hat{a}_{i,t})^2\right]$. The training procedure of Prompt-DT is to autoregressively generate the action conditioned on the current state, return-to-go, past trajectory and sampled prompt.

## 3 The Proposed Framework

Building on the success of LLMs in prompt-based few-shot learning, we propose Language model-initialized Prompt Decision Transformer (LPDT), a novel and effective framework to incorporate powerful pre-trained language models into Decision Transformers to improve their few-shot learning abilities. We further introduce prompt regularization during fine-tuning to better identify tasks. Furthermore, our design is flexible and easy to adapt to other sequence modeling methods for offline RL such as Reinformer (Zhuang et al., 2024). Figure 1 illustrates the overview of our method. At a high level, LPDT incorporates several key components:

- **Language model initialization for Prompt-DT:** We first use a pre-trained language model as the initialization for our Decision Transformer. This design ensures compatibility with the Prompt-DT paradigm, where prompts from various tasks are appended to the input sequence.
- **Parameter-efficient fine-tuning on RL tasks:** We adopt Low-Rank Adaptation (LoRA) (Hu et al., 2021) to fine-tune a low-rank residual matrix while keeping the original weight matrix of the language model fixed throughout the learning process. This approach significantly reduces the number of parameters compared to standard full fine-tuning of the large language models.
- **Prompt regularization through supervised and contrastive objectives:** We incorporate additional regularization over the prompt embeddings to better identify tasks. Specifically, we employ loss functions derived from both supervised and contrastive learning techniques to fully utilize task-related information from prompts, thereby preventing the language model from overfitting specific tasks.
- **Flexiblity of LPDT :** We further extend our proposed framework to other sequence modeling methods for offline RL to demonstrate its flexibility. In particular, we adapt it to Reinformer-based models, incorporating regression on actions and expectile regression on returns to enhance trajectory stitching with maximum-return estimation in offline RL. We demonstrate that language-model initialization benefits Transformer-based prompt sequence modeling methods.

We discuss these techniques in detail in the rest of this section. At the end of this section, built on these components, we present our learning algorithm in Algorithm 1.

### 3.1 Language Model Initialization

We first use a pretrained large language model (LLM) with weights denoted by $M_{\theta^*}$ as the initialization of LPDT. Recent advances in large language models have demonstrated strong few-shot learning abilities. With task-specific information such as prompts for translation or answering questions, language models can generate task-specific responses. We adapt these language models to RL domains such as MuJoCo control tasks, providing prior knowledge that may be relevant to downstream tasks. However, this is not immediately straightforward since LLMs take word tokens as input, which differ from the inputs used in RL. The common next-token prediction objective for the language model (LM) can be formulated as

$$\mathcal{L}_{LM} = \sum_{i=1}^{t-1} -\log(M_{\theta^*}(x_{i+1} \mid x_1, \ldots, x_i)), \tag{3.1}$$

where $M_{\theta^*}$ is the language model, $\theta^*$ is the set of parameters of the pretrained language model and $x_i$ represents the word token. Concretely, we decompose the pretrained parameters as $\theta^* = \{E_{\text{token}}^*, T^*, W_{\text{vocab}}^*\}$, where $E_{\text{token}}^*$ is the token embedding layer, $T^*$ is the causal Transformer backbone, and $W_{\text{vocab}}^*$ is the language model output head. To make $M_{\theta^*}$ compatible with RL trajectories, we keep the causal Transformer backbone and *replace the word embedding layer and language model output head with task-specific linear layers*: an input projection $E_{\text{RL}} \in \mathbb{R}^{d_z \times d_{\text{input}}}$ that maps each RL "token", i.e., $z_t \triangleq [R_t, s_t, a_t]$, into a vector in $\mathbb{R}^{d_{\text{input}}}$ as the input for the Transformer backbone, and an output head $W_{\text{act}} \in \mathbb{R}^{d_{\text{model}} \times d_a}$ that maps the hidden state with dimension $d_{\text{model}}$ to a probability distribution over the action space. Since the original token embedding and output head are trained on the text corpus, they cannot be adapted to RL trajectories. We therefore train our own input projection layer $E_{RL}$ and output head $W_{act}$ from scratch on the RL trajectories dataset. To be more specific, the input projection layer $E_{RL}$ and output head $W_{act}$ are randomly initialized with the frozen pretrained language model backbone. Then the input projection and output head are fully adapted to RL task trajectories and output the action for RL problems. We denote $\theta = \{E_{\text{RL}}, T^*, W_{\text{act}}\}$ as the adapted parameters, and $M_\theta$ is the initialized model. Given a prompt-augmented input sequence $\tau_i^{\text{input}} = [\tau_i^*, \tau_i]$, the model produces action predictions autoregressively under the same causal mask as language modeling:

$$\hat{a}_{i,t} = M_\theta(\tau_{i,1:t}^{\text{input}}) = W_{\text{act}} \cdot T^*\big(E_{\text{RL}}(\tau_{i,1:t}^{\text{input}})\big). \tag{3.2}$$

For continuous control settings (e.g., MuJoCo), we replace the cross-entropy MLE objective in (3.1) with an MSE regression objective by replacing the language model softmax head with the linear regression action head $W_{\text{act}}$. We then optimize the standard Prompt-DT regression loss

$$\mathcal{L}_{\text{PDT}} = \mathbb{E}_{\tau_i^{\text{input}}}\big[\tfrac{1}{K}\sum_{t=1}^{K}\|a_{i,t} - \hat{a}_{i,t}\|_2^2\big]. \tag{3.3}$$

By leveraging the pretrained language model, we preserve broad sequence-modeling priors while adapting only lightweight RL-specific projections.

### 3.2 Parameter-efficient Fine-tuning on RL Tasks

To efficiently adapt the frozen pretrained language model to downstream RL tasks, rather than updating all weights, we apply LoRA (Hu et al., 2021) by introducing trainable low-rank matrices into each target weight matrix while keeping the original backbone fixed. This significantly reduces the number of trainable parameters and enables scalable adaptation to large language models. To be specific, assume the model has $L$ layers. In each layer $l \in \{1, \ldots, L\}$, we update the attention projections $\{W_q^{(l)}, W_k^{(l)}, W_v^{(l)}\}$ which are query, key, and value projections using LoRA. As an example, for the query projection $W_q^{(l)} \in \mathbb{R}^{d_{\text{model}} \times d_q}$ where $d_{\text{model}}$ is the model dimension and $d_q$ is the query dimension, we reparameterize

$$W_q^{(l)} = W_{q,0}^{(l)} + \Delta W_q^{(l)} = W_{q,0}^{(l)} + \alpha_q/r_q A_q^{(l)} B_q^{(l)}, \tag{3.4}$$

where $W_{q,0}^{(l)}$ is the frozen weight inherited from the language model, $\alpha_q$ is the scaling hyperparameter, $r_q$ is the rank in LoRA, and $A_q^{(l)} \in \mathbb{R}^{d_{\text{model}} \times r_q}$, $B_q^{(l)} \in \mathbb{R}^{r_q \times d_q}$ are the trainable low-rank factor matrices with $r_q \ll \min\{d_{\text{model}}, d_q\}$. We apply similar updates to $W_k^{(l)}$ and $W_v^{(l)}$. For simplicity, we denote LoRA factor matrices in the query, key, and value projections as $\{A_{q,k,v}^{(l)}, B_{q,k,v}^{(l)}\}$. By learning only $\Delta W_q^{(l)}$, $\Delta W_k^{(l)}$, and $\Delta W_v^{(l)}$, our method avoids full fine-tuning of the Transformer backbone parameters, substantially reducing compute and memory costs while preserving the generalization ability of the pretrained backbone.

### 3.3 Prompt Regularization with Supervised and Contrastive Objectives

When different tasks are close to each other, naively using the prompts can yield embeddings that are non-discriminative across tasks. Making prompts distinguishable at test time is costly. Previous works such as Prompt-Tuning DT (Hu et al., 2023) and Prompt Diffuser (Hu et al., 2024) aim to tune prompts at test time on unseen tasks. These methods perform additional gradient updates during test-time adaptation, which incurs extra computational cost when deploying policies. To avoid the computational overhead of test-time prompt tuning and improve generalization to unseen tasks, we introduce a task-aware training objective. Specifically, we incorporate a regularization term on the prompt embeddings, termed *prompt regularization*, to encourage the model to learn discriminative prompt representations that reflect task identity during training. Let the prompt encoder be $g_\phi$ and define the prompt embedding $\mathbf{z}_i = g_\phi(\tau_i^*)$.

We use the loss function for Prompt-DT defined in (3.3) as the base loss function, and then incorporate a prompt regularization term. The final loss function of our method is given by

$$\mathcal{L}_{\text{total}} = \mathbb{E}_{\tau_i^{\text{input}} \sim T_i} \Big[ \tfrac{1}{K} \sum_{t=1}^{K} (a_{i,t} - \hat{a}_{i,t})^2 \Big] + \lambda \mathcal{L}_\phi, \tag{3.5}$$

where $\mathcal{L}_\phi$ is the loss for the prompt regularization, which we will specify in the rest of this section, and $\lambda$ is the hyperparameter for prompt regularization. The encoder $g_\phi$ is randomly initialized and jointly trained by (3.5) from scratch.

In particular, we propose two practical implementations of prompt regularization based on supervised learning and contrastive learning methods respectively.

**Supervised learning-based prompt regularization.** In this approach, we add a classifier head to the output of the prompt encoder. We use the task ID from the dataset as the label to help the prompt encoder learn a more meaningful embedding that can easily distinguish different task environments. To be specific, we attach a classifier $C(\cdot)$ to the prompt encoder output and train it with known task IDs, encouraging $\mathbf{z}_i$ to be linearly separable across environments. We adopt the following cross-entropy loss as the regularizer

$$\mathcal{L}_\phi^{\text{classifier}} = - \sum_i y_i \log(\hat{y}_i), \tag{3.6}$$

where $y_i$ is the task label and $\hat{y}_i = C(\mathbf{z}_i)$ is the predicted probability for task $i$ based on the prompt $\tau i^*$.

**Contrastive learning-based prompt regularization.** Unlike the supervised classifier regularization, the contrastive objective relies on the positive or negative information rather than explicit task labels for each task during optimization. This is achieved by assuming that all data points collected within the same batch originate from the same underlying task. Furthermore, this design also allows the method to be adapted to settings where tasks are continuously collected during training without changing the model structure. From an information theory perspective, the ideal prompt encoder should aim to maximize the mutual information between the prompt representation and the tasks. We use the InfoNCE objective (Oord et al., 2018; Yuan & Lu, 2022) to calculate the loss over the prompt. We formulate $\mathcal{L}_\phi$ as:

$$\mathcal{L}_\phi^{\text{InfoNCE}} = -\mathbb{E} \Bigg[ \log \frac{\exp(\text{sim}(\mathbf{z}_i, \mathbf{z}_i^+/\tau))}{\exp(\text{sim}(\mathbf{z}_i, \mathbf{z}_i^+/\tau)) + \sum_{k=1}^{N} \exp(\text{sim}(\mathbf{z}_i, \mathbf{z}_{i,k}^-)/\tau))} \Bigg], \tag{3.7}$$

where $\mathbf{z}_i$ and $\mathbf{z}_i^+$ form a positive prompt pair corresponding to the same task, while $\mathbf{z}_{i,k}^-$ denotes a negative prompt sampled from different tasks. $N$ is the number of negative samples, and $\tau$ is the temperature hyperparameter that controls the sharpness of the distribution. The function $\text{sim}(\cdot, \cdot)$ computes the similarity between two prompt embeddings in which we use the cosine similarity.

### 3.4 Flexibility of LPDT

Our proposed framework is highly flexible and can be seamlessly integrated with other Transformer-based sequence modeling methods. To demonstrate this adaptability, we extend LPDT to the recently introduced Reinformer (Zhuang et al., 2024) as an illustrative example. Reinformer is a return-conditioned sequence model optimized for maximum-return policy learning, which shows state-of-the-art performance on single-task offline RL benchmarks. We demonstrate that our proposed framework LPDT can be easily adapted to Reinformer with minor adjustments.

---

**Algorithm 1** LPDT: Training and Inference

---

**Require:** Pretrained LLM $M_\theta^*$, dataset $\mathcal{D}$, regularization weight $\lambda$, temperature $\tau$

1: **Training:**
2: Freeze all Transformer layers in $\theta^*$
3: Replace the word embedding with linear layer $E_{\text{RL}}$ and the output layer $W_{\text{act}}$
4: Insert LoRA adapters $\{A_{q,k,v}^{(l)}, B_{q,k,v}^{(l)}\}$ into weight matrices
5: **for** Iteration $= 1, \ldots, E$ **do**
6:    **for all** mini-batch $\mathcal{B} \subset \mathcal{D}$ **do**
7:       Build input sequence $\tau_i^{\text{in}} \leftarrow [\tau_i^*;\ \tau_i]$ for each $\tau_i \in \mathcal{B}$
8:       $\hat{a}_{i,1:K} \leftarrow M_\theta(\tau_i^{\text{in}})$
9:       $\mathcal{L}_{\text{PDT}} \leftarrow 1/|\mathcal{B}|K \sum_{\tau_i \in \mathcal{B}} \sum_{t=1}^{K} \|a_{i,t} - \hat{a}_{i,t}\|^2$
10:      **if** *supervised* regularization **then**
11:        $\mathcal{L}_\phi \leftarrow -1/|\mathcal{B}| \sum_{\tau_i \in \mathcal{B}} y_i \log \hat{y}_i$         ▷Equation (3.6)
12:      **else**
13:        $\mathcal{L}_\phi \leftarrow -1/|\mathcal{B}| \sum_{(\mathbf{z}_i, \mathbf{z}_i^+)} \log \dfrac{\exp(\text{sim}(\mathbf{z}_i, \mathbf{z}_i^+)/\tau)}{\exp(\text{sim}(\mathbf{z}_i, \mathbf{z}_i^+)/\tau) + \sum_k \exp(\text{sim}(\mathbf{z}_i, \mathbf{z}_{i,k}^-)/\tau)}$    ▷Equation (3.7)
14:      **end if**
15:      $\mathcal{L}_{\text{total}} \leftarrow \mathcal{L}_{\text{PDT}} + \lambda \mathcal{L}_\phi$
16:      Update LoRA matrix $\{A_{q,k,v}^{(l)}, B_{q,k,v}^{(l)}\}$ and linear layers $\{E_{\text{RL}}, W_{\text{act}}\}$ via $\nabla_\theta \mathcal{L}_{\text{total}}$
17:    **end for**
18: **end for**
19: **Inference on unseen tasks:**
20: Sample few-shot prompt $\tau_{\text{test}}^*$
21: **for** $t = 1$ **to** horizon **do**
22:    $a_t \leftarrow M_\theta([\tau_{\text{test}}^*;\ \tau_{1:t-1}])$
23: **end for**

---

First, Reinformer differs from DT in the ordering of its input sequence. Specifically, Reinformer structures each trajectory segment as a tuple of state, predicted return, and action. This format allows the model to first predict a return and then condition action generation on both the current state and the predicted return. Accordingly, we define the input trajectory for task $i$ as:

$$\tau_i^{\text{rein-input}} = (\tau_i^*, \tau_i) = \left(s_{i,1}^*, R_{i,1}^*, a_{i,1}^*, \cdots, s_{i,K^*}^*, R_{i,K^*}^*, a_{i,K^*}^*, s_{i,1}, R_{i,1}, a_{i,1}, \cdots, s_{i,K}, R_{i,K}, a_{i,K}\right). \quad (3.8)$$

where $s_{i,t}$, $R_{i,t}$, and $a_{i,t}$ denote the state, return, and action at time step $t$, respectively. To better estimate high-reward trajectories across diverse tasks, we employ the expectile loss for return prediction:

$$\mathcal{L}_{\text{return}} = \mathbb{E}_{\tau_i^{\text{rein-input}} \sim T_i} \left[ \frac{1}{K} \sum_{t=1}^{K} \left| m - \mathbb{1}(\Delta R < 0) \right| \cdot \Delta R^2 \right], \quad (3.9)$$

where $\Delta R = R_{i,t} - \hat{R}_{i,t}$ is the difference between the ground-truth return and the predicted return, and $m \in [0,1]$ is a hyperparameter controlling the weight on overestimation or underestimation errors. In practice, we favor overestimation to encourage optimistic trajectory planning during testing. The training loss in Algorithm 1 is then augmented with an additional return loss term as follows:

$$\mathcal{L}_{\text{total}} = \mathcal{L}_{\text{PDT}} + \lambda \mathcal{L}_\phi + \eta \mathcal{L}_{\text{return}}, \quad (3.10)$$

where $\eta$ is the hyperparameter weight for $\mathcal{L}_{\text{return}}$. During inference, the return-first sequence structure is preserved. The model begins by encoding the few-shot prompt from a new task and initializing with the current state. It then predicts a return, followed by generating the corresponding action. This autoregressive rollout continues step-by-step, with the prompt providing task-specific conditioning to ensure alignment with the environment dynamics.

# 4 Experiments

In this section, we conduct experiments to evaluate the few-shot generalization ability of our proposed framework LPDT. We evaluate the performance of LPDT on MuJoCo control tasks (Fu et al., 2020) and Meta-World (Yu et al., 2020) with the episode accumulated reward as the evaluation metric. We also evaluate the prompting ability of LPDT under limited data settings. Our experiments aim to answer the following questions: (1) Can LPDT with language model initialization achieve better performance compared with Prompt-DT and other baselines? (2) Does the language-initialized Transformer model contain the knowledge of the unseen RL tasks and help improve the performance under limited data? (3) Does prompt regularization help the model distinguish different tasks and enhance the prompt learning capability of LPDT? (4) Does the improved performance stem from the language model's better understanding of RL, or from its ability to provide a generally better initialization for learning RL tasks? (5) Can LPDT be adapted to different pre-trained language models other than GPT-2? (6) Can LPDT be extended to sequence models beyond Decision Transformer and Reinformer? (7) Can prompt selection strategies further improve performance?

## 4.1 Implementation

In the empirical study, we first implement our LPDT method with GPT-2 as the language model initialization. GPT-2 is pre-trained on OpenWebtext (Puri & Catanzaro, 2019). During the fine-tuning stage, we follow the same hyperparameters for Prompt-DT (see Appendix D for details). We also leverage LoRA to significantly reduce the number of trainable parameters. For the prompt regularization, we use an MLP to further encode the prompt embedding. For the supervised version of prompt regularization defined in (3.6), we directly use the logits from the MLP to compute the cross-entropy loss and refer to the method as LPDT-Classifier. For the contrastive version of prompt regularization defined in (3.7), we calculate the similarity matrix through the cosine similarity based on the logits from the MLP and refer to it as LPDT-InfoNCE. We also extend our methods to another sequence modeling method adapted from Reinformer (Zhuang et al., 2024) named LPDT-Rein-Classifier and LPDT-Rein-InfoNCE.

## 4.2 Datasets and Baselines

In this work, we evaluate the performance of our proposed approach on MuJoCo control tasks and Meta-World, which are commonly used in existing Prompt-DT type of methods (Xu et al., 2022; Hu et al., 2023; 2024; Wang et al., 2024b), namely, Cheetah-dir, Cheetah-vel, Ant-dir, Point-robot, Meta-World reach-v2.

In Cheetah-dir, there are two tasks with goal directions as forward and backward, where the reward function promotes high velocity along the goal direction. The training and testing phases both include the two tasks. Similar to Cheetah-dir, Ant-dir also segments the tasks by directions. There are 50 tasks in Ant-dir with different goal directions uniformly sampled in 2D space. The tasks are split into 45 training tasks and 5 testing tasks. The ant is also rewarded with high velocity along the goal direction. Different from segmenting the tasks by direction, Cheetah-vel penalizes the agent through the $l_2$ errors with the target velocities sampled from the velocity interval. There are 40 tasks with different goal velocities where 35 tasks are training tasks and 5 tasks are testing tasks. Point-robot is a point navigating to the given goal position. In addition to the MuJoCo control meta-RL tasks, we also test our approach on Meta-World (Yu et al., 2020) which is an open benchmark for meta-RL and multi-task learning. In this work, we evaluate our approach on Meta-World reach-v2. The objective of reach-v2 is to control the robot to reach the target position in 3D positions. Each task has a different goal position.

Specifically, we collect the data by first training the Soft Actor-Critic (Haarnoja et al., 2018) in these different RL task environments and then extracting the data from the replay buffer. More details about the datasets can be found in Appendix C. We compare the few-shot generalization ability of our proposed LPDT with baseline algorithms. For each method, we evaluate performance using the accumulated reward. The baselines we choose include Prompt-DT (Xu et al., 2022), Prompt-Rein, CORRO (Yuan & Lu, 2022), CSRO (Gao et al., 2024) and Meta-DT (Wang et al., 2024b).

## 4.3 Results of LPDT and Baselines

In this section, we conduct experiments on our proposed LPDT framework and baseline methods to evaluate their performance. Additionally, we compare variants of LPDT that use different prompt regularization strategies and sequence modeling techniques. The average accumulated reward per episode in the test task set is used as the evaluation metric for all methods. The results for Prompt-DT, CORRO, CSRO, and Meta-DT are taken from the Meta-DT paper (Wang et al., 2024b), while the results for MW Reach-v2 are provided by our reproduction. We compare our approaches including both the supervised classifier version and the contrastive InfoNCE version with prior works. The model is tested with three different random seeds, and we report the average return across all testing tasks.

Table 1 illustrates that LPDT outperforms baseline algorithms (Prompt-DT, Prompt-Rein, CORRO, and CSRO) on MuJoCo Control tasks and MW Reach-v2 and is competitive with Meta-DT. Specifically, our LPDT approaches outperform Meta-DT in Cheetah-dir and MW Reach-v2. In all other environments, LPDT achieves the second-best performance among the baselines and is competitive compared with Meta-DT. We conclude that using our LPDT approach improves the performance compared to most baselines and is competitive with Meta-DT. It is worth noting that Meta-DT introduces an additional prompt selection stage at test time to select high-quality prompts, while our methods avoid this computation overhead. We use the prompt regularization module to distinguish the prompt instead of iterating over the demonstration set to choose the best prompt. Nevertheless, the prompt selection strategy is orthogonal to our contribution and can be applied at the cost of extra computation.

Table 1: Results for MuJoCo control tasks and MW tasks. The best mean scores are highlighted in green and the second mean scores are highlighted in blue. For each environment, the length of the prompt is $K^* = 5$. The dataset we utilized is the full dataset. We test all the results on unseen tasks with three random seeds. LPDT outperforms baselines on the Cheetah-dir and MW Reach-v2 environment and are competitive remaining environments.

| Method | Cheetah-dir | Cheetah-vel | Ant-dir | Point-robot | MW Reach-v2 |
|---|---|---|---|---|---|
| Prompt-DT | 960.32 ± 17.07 | -133.78 ± 18.24 | 678.07 ± 68.74 | -7.99 ± 0.46 | 2906.67 ± 33.16 |
| Prompt-Rein | 991.10 ± 30.59 | -114.06 ± 22.51 | 702.24 ± 36.89 | -7.77 ± 0.90 | 3008.25 ± 77.81 |
| CORRO | 628.64 ± 61.11 | -111.47 ± 36.97 | 381.42 ± 13.83 | -7.76 ± 0.18 | - |
| CSRO | 641.05 ± 129.54 | -129.00 ± 24.24 | 417.37 ± 39.70 | -19.42 ± 2.10 | - |
| Meta-DT | 874.91 ± 73.45 | -52.42 ± 8.11 | 961.27 ± 18.07 | -6.90 ± 0.11 | 1418.39 ± 6.01 |
| LPDT-Classifier | 1121.68 ± 25.13 | -55.15 ± 15.82 | 782.25 ± 30.39 | -8.19 ± 0.46 | 2903.43 ± 55.09 |
| LPDT-InfoNCE | 1118.95 ± 35.14 | -57.56 ± 9.55 | 775.79 ± 35.28 | -9.06 ± 0.22 | 2946.88 ± 92.18 |
| LPDT-Rein-Classifier | 1131.94 ± 51.09 | -95.02 ± 10.58 | 769.52 ± 69.40 | -7.30 ± 0.35 | 3023.38 ± 178.49 |
| LPDT-Rein-InfoNCE | 1113.96 ± 20.52 | -96.07 ± 5.33 | 759.54 ± 10.95 | -7.37 ± 0.27 | 2614.41 ± 27.57 |
| Mean of Variants | 1121.63 | -75.95 | 771.78 | -7.98 | 2872.03 |

## 4.4 Data Efficiency of LPDT

In this section, we demonstrate the data efficiency of LPDT. The data efficiency shows how well our approaches and baselines can perform with access to only a certain number of unique trajectories after sufficient training iterations. The pretrained language model provides prior knowledge for reinforcement learning tasks, leading to increased data efficiency within our framework. To validate this hypothesis, we conduct extensive experiments on various datasets split by ratios $\{1.0, 0.5, 0.1, 0.05, 0.01\}$. A ratio of 1.0 corresponds to the original full dataset, while a ratio of 0.01 represents an extreme scenario where only 1% of the original full dataset is available in training. We train LPDT and compare it with Prompt-DT and Meta-DT, under these conditions and evaluate their performance on unseen tasks. The results are presented in Table 2.

As depicted in Table 2, the return score decreases as the dataset size diminishes. In tasks such as Cheetah-dir and Ant-dir, our LPDT algorithm mostly achieves better performance compared to Prompt-DT and Meta-DT. Specifically, on Cheetah-dir, LPDT is consistently the best across all splits from 1.0 to 0.05. On Cheetah-vel, Ant-dir and Point-robot, with full dataset, the performance of LPDT is comparable to Meta-DT. But when it decreases to a small dataset with 0.5, 0.1, 0.05, our LPDT outperforms the baseline. With

only 10% of the dataset, our proposed methods can achieve similar or even better performance compared to Prompt-DT in Cheetah-vel and Ant-dir. Nevertheless, at a ratio of 0.01, an extreme case, all methods perform poorly on Cheetah-dir and the scores are mostly determined by noise, which is expected due to insufficient training data. For the Cheetah-vel and MW Reach-v2 tasks, results are inconsistent and exhibit significant variance, likely due to inherent task challenges and difficulties in learning stable policies in these environments. Meta-DT requires learning a context encoder based on training trajectories and is particularly sensitive to limited data availability. Despite these challenges, the overall trend indicates that **LPDT models maintain a performance advantage over other baselines, especially when dataset sizes are moderately reduced**. This underscores the benefit of leveraging prior knowledge from language models to enhance data efficiency in downstream reinforcement learning tasks.

Table 2: Results for MuJoCo control tasks and MW tasks with the $\{0.01, 0.05, 0.1, 0.5, 1.0\}$ ratio dataset. The best mean scores are highlighted in green and the second mean scores are highlighted in blue. For each environment, the length of the prompt is $K^* = 5$. We test all the results on unseen tasks with three random seeds. We demonstrate that language initialization can improve the performance of LPDT.

| Dataset Ratio | Algorithm | Cheetah-dir | Cheetah-vel | Ant-dir | Point-robot | MW Reach-v2 |
|---|---|---|---|---|---|---|
| | Prompt-DT | 960.32 ± 17.07 | -133.78 ± 18.24 | 678.07 ± 68.74 | -7.99 ± 0.46 | 2906.67 ± 33.16 |
| | Meta-DT | 874.91 ± 73.45 | **-52.42 ± 8.11** | **961.27 ± 18.07** | **-6.90 ± 0.11** | 1418.39 ± 6.01 |
| 1.0 | LPDT-Classifier | **1121.68 ± 25.13** | -55.15 ± 15.83 | 782.25 ± 30.39 | -8.19 ± 0.46 | 2903.43 ± 55.09 |
| | LPDT-InfoNCE | 1118.95 ± 35.14 | -57.56 ± 9.55 | 775.79 ± 35.28 | -9.06 ± 0.22 | **2946.88 ± 92.18** |
| | LPDT-Rein-Classifier | **1131.94 ± 51.09** | -95.02 ± 10.58 | 769.52 ± 69.40 | **-7.30 ± 0.35** | **3023.38 ± 178.49** |
| | LPDT-Rein-InfoNCE | 1113.96 ± 20.52 | -96.07 ± 5.33 | 759.54 ± 10.95 | -7.37 ± 0.27 | 2614.41 ± 27.57 |
| | Mean of Variants | 1121.63 | -75.95 | 771.78 | -7.98 | 2872.03 |
| | Prompt-DT | 691.05 ± 72.87 | -69.93 ± 7.59 | 663.49 ± 17.06 | **-6.53 ± 0.41** | **3062.35 ± 15.89** |
| | Meta-DT | 202.82 ± 33.91 | -75.61 ± 7.77 | 508.73 ± 61.73 | -14.07 ± 0.26 | 2085.79 ± 2.62 |
| 0.5 | LPDT-Classifier | **1074.20 ± 27.26** | **-55.19 ± 10.94** | 705.99 ± 51.27 | **-6.04 ± 0.56** | 2426.50 ± 125.30 |
| | LPDT-InfoNCE | **1102.18 ± 43.18** | **-64.23 ± 5.76** | 714.31 ± 67.15 | -6.97 ± 1.55 | 2439.34 ± 131.92 |
| | LPDT-Rein-Classifier | 949.85 ± 25.68 | -101.84 ± 4.23 | **746.45 ± 56.32** | -7.832 ± 0.94 | 2841.04 ± 136.94 |
| | LPDT-Rein-InfoNCE | 899.25 ± 32.85 | -106.94 ± 8.52 | **750.38 ± 70.14** | -7.468 ± 0.44 | **2902.60 ± 92.70** |
| | Mean of Variants | 1006.37 | -82.05 | 729.28 | -7.08 | 2652.37 |
| | Prompt-DT | 193.69 ± 9.06 | **-72.47 ± 3.14** | 598.29 ± 21.61 | -8.94 ± 0.43 | **3541.67 ± 75.44** |
| | Meta-DT | 77.42 ± 11.90 | -114.32 ± 10.24 | 509.25 ± 51.03 | -16.68 ± 1.64 | 2040.93 ± 166.34 |
| 0.1 | LPDT-Classifier | **750.90 ± 88.86** | **-65.23 ± 19.16** | **688.21 ± 36.78** | **-8.46 ± 0.34** | 2519.92 ± 152.28 |
| | LPDT-InfoNCE | **666.13 ± 56.85** | -77.08 ± 17.71 | 660.63 ± 52.00 | **-8.19 ± 0.91** | 2577.42 ± 60.06 |
| | LPDT-Rein-Classifier | 622.95 ± 45.98 | -107.29 ± 15.12 | **687.71 ± 68.32** | -9.242 ± 0.54 | 2781.77 ± 170.80 |
| | LPDT-Rein-InfoNCE | 614.52 ± 85.21 | -112.69 ± 14.25 | 641.87 ± 45.96 | -9.609 ± 1.23 | **2827.77 ± 154.02** |
| | Mean of Variants | 663.63 | -90.57 | 669.61 | -8.88 | 2676.72 |
| | Prompt-DT | 112.40 ± 14.43 | **-97.29 ± 9.90** | **491.320 ± 19.94** | -8.26 ± 0.07 | **3320.88 ± 77.59** |
| | Meta-DT | 37.87 ± 19.15 | -118.15 ± 13.35 | 309.68 ± 63.76 | -18.27 ± 1.67 | 2004.85 ± 44.59 |
| 0.05 | LPDT-Classifier | **386.01 ± 84.45** | -103.43 ± 5.45 | 488.54 ± 43.80 | **-7.80 ± 1.22** | 2510.56 ± 120.62 |
| | LPDT-InfoNCE | **320.28 ± 79.24** | **-102.5 ± 5.09** | 456.47 ± 65.82 | -8.32 ± 0.45 | **2592.62 ± 153.21** |
| | LPDT-Rein-Classifier | 252.33 ± 69.15 | -113.70 ± 6.33 | **534.28 ± 52.31** | -10.22 ± 1.23 | 2508.67 ± 124.36 |
| | LPDT-Rein-InfoNCE | 279.48 ± 45.58 | -112.68 ± 5.48 | 458.98 ± 43.64 | -10.50 ± 1.44 | 2295.21 ± 91.08 |
| | Mean of Variants | 309.53 | -108.08 | 484.57 | -9.21 | 2476.77 |
| | Prompt-DT | 18.97 ± 4.52 | **-99.89 ± 2.85** | 183.55 ± 11.39 | **-8.91 ± 0.41** | **2551.17 ± 122.52** |
| | Meta-DT | **30.88 ± 21.07** | -173.93 ± 28.09 | 90.86 ± 0.91 | -17.28 ± 2.01 | 1934.47 ± 28.61 |
| 0.01 | LPDT-Classifier | 17.80 ± 12.73 | **-121.84 ± 4.12** | 175.82 ± 18.61 | **-10.56 ± 0.21** | 2132.60 ± 165.87 |
| | LPDT-InfoNCE | 10.54 ± 6.75 | -127.81 ± 3.26 | 130.76 ± 3.64 | -11.52 ± 1.80 | 2390.82 ± 79.37 |
| | LPDT-Rein-Classifier | **25.42 ± 14.50** | -125.22 ± 2.58 | **264.61 ± 12.04** | -11.15 ± 0.47 | **2508.67 ± 124.36** |
| | LPDT-Rein-InfoNCE | 9.26 ± 10.32 | -133.44 ± 4.71 | **270.43 ± 8.36** | -10.93 ± 1.12 | 2449.91 ± 91.08 |
| | Mean of Variants | 15.755 | -127.08 | 210.41 | -11.04 | 2370.50 |

## 4.5 Ablation Studies

In this section, we provide ablation studies on our proposed framework LPDT from various aspects. We focus on the effectiveness of prompt regularization and the language model. We also extend our proposed framework to other sequence modeling methods in decision-making. Finally, we demonstrate the different prompt selection strategies during the testing stage.

Table 3: Results for MuJoCo control tasks and MW tasks with different regularization methods of our method including w/o regularization, classifier regularization and InfoNCE regularization. For each environment, the length of the prompt is $K^* = 5$. We test all the results on unseen tasks with three random seeds. The dataset we utilized is the full dataset. We demonstrate that the regularization on prompts can help distinguish the task and improve the performance compared with the method without regularization.

| Task | LPDT | | | | LPDT-Rein | | | |
|---|---|---|---|---|---|---|---|---|
| | w/o regularization | w/ regularization | | | w/o regularization | w/ regularization | | |
| | | Classifier | InfoNCE | Mean | | Classifier | InfoNCE | Mean |
| Cheetah-dir | $1109.54 \pm 20.04$ | $1121.68 \pm 25.13$ | $1118.95 \pm 35.14$ | 1120.32 | $1123.51 \pm 9.67$ | $1131.94 \pm 51.09$ | $1113.96 \pm 20.52$ | 1122.95 |
| Cheetah-vel | $-56.38 \pm 7.53$ | $-55.15 \pm 15.83$ | $-57.56 \pm 9.55$ | -56.4 | $-95.39 \pm 23.80$ | $-95.02 \pm 10.58$ | $-96.07 \pm 5.33$ | -95.55 |
| Ant-dir | $762.19 \pm 59.22$ | $782.25 \pm 30.39$ | $775.79 \pm 35.28$ | 779.02 | $718.78 \pm 43.41$ | $769.52 \pm 69.40$ | $759.54 \pm 10.95$ | 764.53 |
| Point-robot | $-6.37 \pm 0.44$ | $-8.19 \pm 0.46$ | $-9.06 \pm 0.22$ | -8.63 | $-6.12 \pm 0.26$ | $-7.30 \pm 0.35$ | $-7.37 \pm 0.27$ | -7.34 |
| MW reach-v2 | $2808.15 \pm 189.82$ | $2903.43 \pm 55.09$ | $2946.88 \pm 92.18$ | 2925.16 | $2252.68 \pm 156.39$ | $3023.38 \pm 178.49$ | $2614.41 \pm 27.57$ | 2818.90 |

**The role of prompt regularization** We first compare our proposed model with its variants that do not utilize prompt regularization, denoted as LPDT w/o regularization and LPDT-Rein w/o regularization. Table 3 illustrates that our proposed LPDT outperforms LPDT w/o regularization and LPDT-Rein w/o regularization in most of our proposed settings. We further compare the two prompt regularization variants. Both variants improve performance over the variants without regularization across most environments, except for Point-robot. The Classifier variant slightly outperforms InfoNCE while InfoNCE variants adapt better to settings with continuously collected tasks. The InfoNCE variant also demonstrates consistent improvements in most settings compared with the versions without regularization. These results show that **prompt regularization improves the model's ability to distinguish between tasks and demonstrates improvements in most settings.**

**The benefits of using pretrained language models** We aim to answer the question of *how initializing DT with pre-trained language models benefits RL tasks*. Specifically, we aim to determine whether the performance improvement arises from (1) *higher-quality prompt embeddings* produced by the language model, which enhance the contextual representation of RL environments, or (2) *knowledge priors for decision-making tasks* encoded in the language model's hidden layers, which facilitate a better understanding of the underlying environment dynamics. To this end, we conduct experiments where we use the text-pretrained model's prompt embedding as input to a freshly initialized Decision Transformer or Reinformer, denoted as 'Prompt-DT-Text-Embedding' and 'Prompt-DT-Rein-Text-Embedding', respectively. In detail, these two variants adopt a language model to encode the trajectory prompt, and the resulting prompt embedding is then used to train a newly initialized Prompt-DT. Since the language model is only employed to extract trajectory embeddings while the DT/Reinformer parameters are trained from scratch, this ablation study allows us to assess whether the language model merely provides better representations or can also be directly fine-tuned into an effective decision-maker, as in our LPDT methods.

The experimental results are summarized in Table 4. We first observe that Prompt-DT-Text-Embedding and Prompt-DT-Rein-Text-Embedding outperform vanilla Prompt-DT methods across all environments except Cheetah-dir, suggesting that the language model can, to some extent, generate more informative prompt embeddings. Moreover, our LPDT algorithms, which use pre-trained language models not only to initialize the embeddings of prompts and trajectories but also to initialize the intermediate layers of DT/Reinformer, significantly outperform the Text-Embedding baselines. This demonstrates that **initializing DTs with language models provides not only improved representations of RL environments but also strong priors for decision-making**—enabling effective adaptation with only minimal fine-tuning.

**Flexibility in the language model** We implemented various pre-trained language models to demonstrate that our proposed approach can be adapted to different language models. In this section, we provide the results with small language models such as Qwen2.5-0.5B (Qwen et al., 2025). Table 5 presents the Ant-dir results under different pre-trained language model initializations. The results indicate that with Qwen2.5-0.5B, the performance remains competitive in LPDT-Rein and improved in LPDT, **demonstrating the**

Table 4: Results for MuJoCo control tasks on Prompt-DT-Text-Embedding and Prompt-DT-Rein-Text-Embedding. For each environment, the length of the prompt is $K^* = 5$. We test all the results on unseen tasks with three random seeds. The dataset we utilized is the full dataset. We demonstrate that improved performance comes from the initialization of the language model.

| Task | DT-based methods | | | | Reinformer-based methods | | | |
|---|---|---|---|---|---|---|---|---|
| | Prompt-DT | | LPDT | | Prompt-DT | | LPDT | |
| | Vanilla | Text-Embedding | Classifier | InfoNCE | Vanilla | Text-Embedding | Classifier | InfoNCE |
| Cheetah-dir | $960.32 \pm 17.07$ | $-23.74 \pm 18.94$ | $1121.68 \pm 25.13$ | $1118.95 \pm 35.14$ | $991.10 \pm 30.59$ | $1015.397 \pm 50.47$ | $1131.94 \pm 51.09$ | $1113.96 \pm 20.52$ |
| Cheetah-vel | $-133.78 \pm 18.24$ | $-72.28 \pm 8.25$ | $-55.15 \pm 15.82$ | $-57.56 \pm 9.55$ | $-114.06 \pm 22.51$ | $-115.82 \pm 9.24$ | $-95.02 \pm 10.58$ | $-96.07 \pm 5.33$ |
| Ant-dir | $678.07 \pm 68.74$ | $716.62 \pm 44.15$ | $782.25 \pm 30.39$ | $775.79 \pm 35.28$ | $702.24 \pm 36.89$ | $696.00 \pm 46.69$ | $769.52 \pm 69.40$ | $759.54 \pm 10.95$ |

**flexibility of our approach to various language model initializations**. We hypothesize that larger language models such as 7B models are likely to further enhance performance in unseen test settings.

Table 5: Results for Ant-Dir with different initialization of pre-trained language model on the full dataset. The length of the prompt is $K^* = 5$. We test all the results on unseen tasks with three random seeds. We demonstrate that LPDT with Qwen2.5-0.5B has improved performance compared to GPT-2.

| LPDT-Classifier GPT-2 | LPDT-InfoNCE GPT-2 | LPDT-Classifier Qwen2.5-0.5B | LPDT-InfoNCE Qwen2.5-0.5B |
|---|---|---|---|
| $782.25 \pm 30.39$ | $775.79 \pm 35.28$ | $800.22 \pm 55.16$ | $786.16 \pm 15.55$ |
| LPDT-Rein-Classifier GPT-2 | LPDT-Rein-InfoNCE GPT-2 | LPDT-Rein-Classifier Qwen2.5-0.5B | LPDT-Rein-InfoNCE Qwen2.5-0.5B |
| $769.52 \pm 69.40$ | $759.54 \pm 10.95$ | $735.97 \pm 58.33$ | $719.69 \pm 35.33$ |

**Extension to other DT methods**   Our proposed method builds upon the high-level concept of Prompt-DT by leveraging prompts and contextual information to guide models in unseen tasks, but extends beyond this specific architecture. Our framework is a versatile plug-in method based on prompt strategies that can be integrated into various architectures. While our primary implementation follows the Prompt-DT paradigm, we also demonstrate our algorithm's flexibility by adapting it to Reinformer, a recent DT-based method achieving state-of-the-art performance in single-task offline RL problems. To further validate this generalizability, we now provide an additional implementation based on EDT (Wu et al., 2024). Table 6 compares our LPDT-EDT approach to standard Prompt-EDT without language initialization, demonstrating significant performance improvements. **These results highlight our framework's potential and flexibility to enhance a wide range of architectural designs.**

Table 6: Results for MuJoCo control tasks on LPDT-EDT on the full dataset. The best mean scores are highlighted in green and the second mean scores are highlighted in blue. The length of the prompt is $K^* = 5$. We test all the results on unseen tasks with three random seeds. We demonstrate that our proposed framework LPDT can be flexible to the sequence modeling methods in decision making.

| Task | Prompt-EDT | LPDT-EDT-Classifier | LPDT-EDT-InfoNCE |
|---|---|---|---|
| Cheetah-dir | $796.56 \pm 87.09$ | $\mathbf{1040.087 \pm 34.834}$ | $1024.172 \pm 52.357$ |
| Cheetah-vel | $-257.14 \pm 1.42$ | $-254.301 \pm 1.213$ | $\mathbf{-222.05 \pm 11.589}$ |
| Ant-dir | $240.794 \pm 24.34$ | $\mathbf{733.435 \pm 16.969}$ | $615.087 \pm 20.267$ |

**Advanced prompt sampling strategies**   In our implementation, prompt trajectories are randomly sampled from the demonstration set during testing on unseen tasks. This is a simple and straightforward approach that allows us to better highlight the benefits of language initialization and low-rank fine-tuning. We additionally implement a prompt selection method based on online interaction returns, using these returns as a performance metric to identify effective prompts. Specifically, we select several prompts from the demonstration set and then interact with the environment several times to obtain the prompt with the highest mean return. We compare the performance of such methods (with suffix '-select') to LPDT. The results, illustrated in Table 7, demonstrate a performance improvement over random selection. However,

the improvement over the current performance of our LPDT is not significant. **These additional prompt selection strategies introduce additional model complexity with a little improvement.** Thus, we use a simpler way which selects the prompts randomly from the demonstration set to implement our LPDT. Nonetheless, this selection strategy is orthogonal to our core LPDT framework and represents an interesting direction for future exploration.

Table 7: Results for MuJoCo control tasks on LPDT and prompt-selection variants. The best mean scores are highlighted in green and the second mean scores are highlighted in blue. For each environment, the length of the prompt is $K^* = 5$. We test all the results on unseen tasks with three random seeds. The dataset we utilized is the full dataset. We demonstrate that our proposed framework LPDTcan be orthogonal to prompt selection strategies and can be improved with advanced prompt selection methods.

| Task | LPDT-Classifier | LPDT-InfoNCE | LPDT-Classifier-Prompt select | LPDT-InfoNCE-Prompt select |
|---|---|---|---|---|
| Cheetah-dir | $1121.68 \pm 25.13$ | $1118.95 \pm 35.14$ | **$1151.33 \pm 24.36$** | **$1139.11 \pm 16.88$** |
| Cheetah-vel | **$-55.15 \pm 15.82$** | $-57.56 \pm 9.55$ | **$-51.02 \pm 16.03$** | $-56.38 \pm 14.21$ |
| Ant-dir | $782.25 \pm 30.39$ | $775.79 \pm 35.28$ | **$824.79 \pm 46.04$** | **$782.72 \pm 38.68$** |

**Computational analysis** To assess the computation cost, we design specific experiments to measure the number of trainable Transformer parameters, training time, and inference time for our proposed LPDT compared to the baselines. We select the Ant-dir as the instance to analyze. All experiments are conducted on a single NVIDIA A6000 GPU with Intel Xeon Ice Lake Gold 5317 processors. For the training time, we compare the methods by achieving the same performance on unseen test tasks. Since these methods achieve different performance levels on Ant-dir, we choose a target score of approximately 650 for a fair comparison, which corresponds to the performance of Prompt-DT. As shown in Table 8, our proposed LPDT achieves similar performance (around 650) to Prompt-DT, with only slightly higher memory and time costs. Compared to Meta-DT, our LPDT variants achieve faster inference because they avoid the additional overhead introduced by prompt selection. It is also worth noting that Meta-DT needs to train a world model which has a longer training time. In addition, LPDT-Rein requires roughly twice the inference time of LPDT. This is reasonable as the LPDT-rein needs to predict the return and action during the inference.

Table 8: Comparison of computational costs on the Ant-dir environment. Training Time refers to the duration required for each method to achieve comparable performance. Inference Time is the overall time taken to generate one trajectory over five test tasks. The number of trainable parameters for the Transformer component of each model is also reported.

| Method | Transformer Trainable Params (M) | Training Time | Inference Time |
|---|---|---|---|
| Prompt-DT | 0.60 | $\sim$3.8h | 4.96s |
| Meta-DT | 0.79 | $\sim$14.7h | 10.41s |
| LPDT-Classifier | 0.91 | $\sim$5.6h | 7.33s |
| LPDT-InfoNCE | 0.91 | $\sim$5.7h | 7.82s |
| LPDT-Rein-Classifier | 0.91 | $\sim$4.7h | 16.96s |
| LPDT-Rein-InfoNCE | 0.91 | $\sim$4.7h | 18.02s |

## 5 Conclusion

In this work, we proposed a novel flexible framework for improving the few-shot prompt ability of decision transformers and other sequence modeling methods in offline reinforcement learning. This framework, termed Language model-initialized Prompt Decision Transformer (LPDT), leverages pre-trained language models integrated with domain-specific RL datasets to enhance few-shot prompt capabilities. LPDT demonstrates superior or competitive performance compared to existing baselines in terms of cumulative rewards on unseen tasks. Our approach holds significant potential for reducing data requirements in offline RL tasks, thereby

increasing applicability in real-world scenarios where large-scale collection of RL trajectories is challenging. Moreover, our results underscore the importance of utilizing pre-trained language models as a foundation for decision-making tasks and demonstrate the effectiveness of our prompt regularization methods in enhancing task-specific information discernment within prompts. Additionally, our LPDT framework exhibits flexibility to accommodate various pre-trained language model initializations and sequence modeling methodologies, including Reinformer and other recent DT variants.

While LPDT has shown promising results, there are several limitations to our approach. Current computing resource constraints limit our use of language models to GPT-2 and Qwen2.5-0.5B. To fully harness the potential of pre-trained language models in decision-making tasks, future efforts will aim at extending our framework to encompass more extensive open-source language models and implementing efficient fine-tuning techniques. Exploration into alternative architectures or the incorporation of multi-task learning could further enhance LPDT's performance. Future research will focus on addressing these limitations and expanding LPDT's application across a broader range of pre-trained language models and decision-making tasks, offering a promising direction for advancements in offline reinforcement learning.

## Acknowledgments

We would like to thank the anonymous reviewers and the editor for their helpful comments. YY and PX were supported in part by the National Science Foundation (DMS-2323112) and the Whitehead Scholars Program at the Duke University School of Medicine. The views and conclusions in this paper are those of the authors and should not be interpreted as representing any funding agency.

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

## A  Code Availability

All source code and experiment configurations are available at `https://github.com/panxulab/LPDT`.

## B  Related Work

**Decision Transformer.** Decision Transformer (DT) (Chen et al., 2021) emerges as a type of algorithm for offline RL by using the powerful Transformer architecture for decision-making. DT models RL trajectories as a sequence generation problem and utilizes the next-token generation paradigm for training. Thus, DT takes the history tokens such as the return, state, and action to predict the next action, which formulates the decision-making as an action prediction or sequence generation in a supervised fashion. Since DT can fully utilize the whole trajectories and is easy to train compared with dynamic programming-based offline RL, many of the following works improved the performance under different settings. For example, Lee et al. (2022) proposes to use the Multi-game Decision Transformer which is trained on part of the Atari games as the multi-tasks training and fine-tuned on the remaining games to achieve efficient adaption. Hyper Decision Transformer (Xu et al., 2023) adds an adapter into DT and is fine-tuned on unseen tasks through the demonstration without expert actions. Xie et al. (2023) proposes to predict the action conditioned on the future trajectory embedding as the hindsight instead of conditioned on the return. Trajectory Transformer (Janner et al., 2021) is another research line, which is trained on sequences of state, action, and rewards and generated with the beam search. Elastic DT (Wu et al., 2024) introduces the expectile regression over the returns and uses the different lengths of context to predict the next action. Reinformer (Zhuang et al., 2024) utilizes the expectile regression to max the return to improve the stitching ability during the sequence modeling. DT can be considered as a special instance of return-conditioned supervised learning, and the theoretical capability has been discussed in Brandfonbrener et al. (2022) and Liu et al. (2025). It has also been applied to off-dynamics RL settings to address policy transfer problems (Wang et al., 2024a).

**Prompt-DT.** Prompt DT (Xu et al., 2022) utilizes the prompt-based framework to do the meta-RL. It is trained on multi-RL tasks with offline datasets. During the training, the prompts or demonstrations which are a small part of the trajectory are combined with trajectories. During the testing on unseen tasks, the prompt can be a guide for indicating the tasks and help the model predict the action to interact with the environments. Following Prompt-DT, several works are adopting the prompt tuning method to achieve a high-quality prompt. Prompt-Tuning DT (Hu et al., 2021) uses the preference ranking function and black-box tuning method to tune the prompt when testing on unseen tasks to achieve a high-quality prompt. Moreover, Prompt Diffuser (Hu et al., 2024) leverages the diffusion model to generate high-quality prompts leading to improved performance in downstream RL tasks. Different from these works, we adopt the prompt regularization which aims to learn a high-quality prompt embedding to distinguish the different but similar RL tasks. Our method adopts this regularization during the training procedure in the prompt dataset. Meta-DT (Wang et al., 2024b) proposes to pre-train a context encoder to encode the context in trajectories. It uses the trajectory context as the input to incorporate the task representation into Prompt-DT.

**Language model based DT.** Large language models have achieved many surprising effects in various tasks in recent years. Pre-trained on large datasets such as the corpus of the Internet, LLMs such as GPTs (Radford et al., 2019) demonstrate prompt ability which can generate the text with the guide of the task information. The success of the large language models motivates the increasing use of pre-trained language models in improving Decision Transformer to solve RL tasks (Chen et al., 2021). Several works utilize the powerful representation generalization ability of language models as policies to do the decision-making. Li et al. (2022) proposed to adopt the pre-trained language models for interactive decision-making to convert the policies to sequence data. Wik-RL (Reid et al., 2022) uses a pre-trained language model from the next-token generation paradigm as the initialization of DT for offline RL tasks. However, it suffers from inferior performance than directly using DT. To overcome these challenges and unleash the power of language models, Shi et al. (2024) proposed the LaMo algorithm which uses a pre-trained language model and parameter-efficient fine-tuning methods to improve the original DT. Zhang et al. (2024) also proposed to use LaMo in partially observable continuous control problems which demonstrates a strong generalization ability. All these methods are designed for single tasks. DPDT (Zheng et al., 2024) uses the pre-trained language model to provide the prior knowledge to train the prompt and utilizes the test time adaptation to

align the cross-task prompts in unseen tasks. However, it still needs a test time adaption phase to achieve the ideal performance. Our approach is fine-tuned for learning to identify different prompts for various RL tasks. And during the testing phase, just a small part of the trajectories is used in our method as the prompt without updating the model.

## C  Details on the Experiment Environments

We evaluate our approach over MuJoCo control tasks and Meta-World ML1 tasks. We split the tasks in these environments into the training set and the testing set. The tasks in Cheetah-dir and Ant-dir are split by directions. The tasks in Cheetah-vel are split by the goal velocities. The tasks in Point-robot are split by the goal positions which are uniformly distributed in a unit square. In Meta-World, the tasks are defined by different goal positions. The detailed task indexes can be found in Table 9. The experiments we conducted are all followed to this setting which guarantees consistency during the evaluation.

Table 9: Training and testing task indexes when testing the generalization ability. We follow the tasks split between Prompt-DT and previous works to guarantee a direct comparison with baselines.

| Environment | Number of tasks | Tasks indexes |
|---|---|---|
| Cheetah-dir | Training set: 2 | [0,1] |
|  | Testing set: 2 | [0,1] |
| Cheetah-vel | Training set: 45 | [0-44] |
|  | Testing set: 5 | [45-49] |
| Ant-dir | Training set: 45 | [0-44] |
|  | Testing set: 5 | [45-49] |
| Point-robot | Training set: 45 | [0-44] |
|  | Testing set: 5 | [45-49] |
| Meta-World reach-v2 | Training set: 15 | [0-14] |
|  | Testing set: 5 | [15-19] |

## D  Hyperparameters

In this section, we show the hyperparameters of our LPDT conducted in Table 1. The hyperparameters have two parts which are the hyperparameters around the transformer and prompt regularization. We list these hyperparameters in Table 10.

## E  Training Return Curves

Figure 2 presents the evaluation of Prompt-DT and our four LPDT approaches. All the plotted methods are tested through 50 episode returns on the unseen tasks. The tasks Cheetah-dir, Cheetah-vel, and Ant-dir have prompts of length $K^* = 5$. Figure 2 shows that they need fewer sample data compared with Prompt-DT to achieve superior performance. The benefit of introducing the language model is to improve the data efficiency which is important in real-world application.

Table 10: Detail on hyperparameters used in our experiments in Table 1. We show that the hyperparameters in two parts which are parameters for model backbone and prompt regularization respectively.

| Hyperparameters | Value |
|---|---|
| Length of training $\tau$ | 20 |
| Length of prompt $K$ | 5 |
| Training batch size for each task | 8 |
| Number of evaluation episodes for each task | 20 |
| Learning rate | 1e-4 |
| Learning rate decay weight | 1e-4 |
| Language initialization | GPT-2 |
| Embedding dimension | 128 |
| Activation | ReLU |
| Classifier hyperparameter | 0.1 |
| Classifier layers | 2 |
| Classifier MLP dimension | 128 |
| InfoNCE hyperparameter | 0.1 |
| InfoNCE temperature | 1 |
| InfoNCE MLP dimension | 128 |

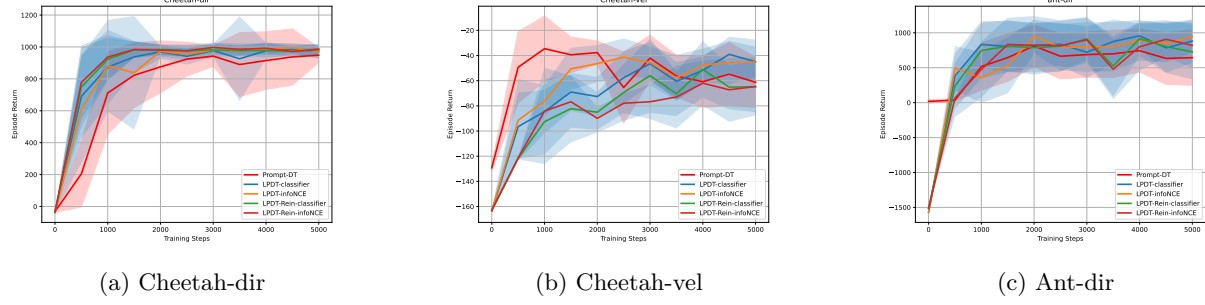

(a) Cheetah-dir          (b) Cheetah-vel          (c) Ant-dir

Figure 2: Training curves on MuJoCo controls with three tasks: Cheetah-dir, Cheetah-vel and Ant-dir for Prompt-DT and our four methods LPDT-Classifier, LPDT-InforNCE, LPDT-Rein-Classifier and LPDT-Rein-InforNCE. The dataset we utilized is the full dataset. We plot the figures on one unseen task with the average returns over 20 evaluation episodes. The figures demonstrate that our LPDT needs less sample data to achieve good performance and be more stable during the training.

