# OpenReview forum: "Pre-trained Language Models Improve the Few-shot  Prompt Ability of Decision Transformer"
_TMLR — Accepted by TMLR_

### Review · Reviewer_6SHM · 2025-09-14

**Summary Of Contributions:**

Contributions:

The authors propose Language model-initialized Prompt Decision Transformer (LPDT), a RL framework that incorporates pre-trained language models.
The authors use Low-Rank Adaptation (LoRA) and prompt regularization methods to efficiently fine-tune the model and enhance the model's ability to distinguish tasks.
The authors also demonstrate the generalizability by extending the framework to a recent DT variant Reinformer.

Strength:

1. The authors have conducted comprehensive experiments and ablations to show the effectiveness of the proposed method.
2. The paper is well-written and easy to follow.

Weakness:

1. The improvement over baselines (Meta-DT) is not obvious (See Table 1). Meta-DT remains more competitive in three environment.
2. Tables are not well-organized. The color and bold criterion are not consistent. Different LPDT variants are shown but no specific discussion over different variants.

**Audience:**

Yes

**Audience Explanation:**

This submission is valuable to the RL community for the people working on decision transformer. It brings a new direction to incorporate pre-trained language models into prompt decision transformer and introduces insights to further research.

**Claims And Evidence:**

Yes

**Claims Explanation:**

The authors conduct comprehensive experiments and ablation studies to validate the proposed claims.

**Requested Changes:**

1. As LPDT-Rein variants results are reported in Table 1, it is better to report the results of Reinformer to compare with.
2. Table 3 mainly shows the ablation results, so it is unnecessary to show Prompt-DT results. It's better to re-organize the table so that the comparison between w/ and w/o regularization is more obvious.
3. As mentioned in the weakness, the authors should consider make the color and bold criterion the same across all tables. Currently in Table 1 the best two results are colored; in table 2 the best results are bold while in other tables no values are colored or bolded.
4. The authors report the values for different variants (Classifier/InfoNCE). It would be better if the authors could provide additional analysis on the performance of these two variants.

---

> ### Author Response · Authors · 2025-10-16
>
> We thank the reviewer for their valuable time and effort in providing detailed feedback on our work. We have uploaded a revised version with changes highlighted in blue text. We added the experiments on Prompt-Rein and reorganized tables for better readability. We also revised several unclear statements in Section 3.1, Section 3.3 and Section 4.5. In the following, we provide a detailed response to your questions and hope you find them satisfying.
>
>
> ---
>
> >**Q1**: The improvement over baselines (Meta-DT) is not obvious (See Table 1). Meta-DT remains more competitive in three environments.
>
> **A1**: As stated in the experiments section and shown in Table 1, our LPDT approach outperforms most baselines and is competitive with Meta-DT. We would like to clarify that Meta-DT introduces an additional prompt selection stage at test time, which incurs computational overhead. However, our methods avoid this computation overhead. We have demonstrated this in the table presented in our response to **Q2** of Reviewer 6SHM. In addition to this, in Table 2 of our manuscript, we demonstrate that our proposed LPDT outperforms the Meta-DT as the size of the dataset decreases. In summary, our LPDT framework achieves performance competitive with Meta-DT and shows superior results in limited-data scenarios.
>
> ---
> >**Q2**: As LPDT-Rein variants results are reported in Table 1, it is better to report the results of Reinformer to compare with.
>
> **A2**: Thank you for the helpful suggestion. We have added the results of the Prompt-Rein baseline to enable a direct comparison with our LPDT-Rein variants. As shown in the table below, our proposed framework consistently outperforms the baseline across multiple environments, demonstrating that LPDT can be seamlessly adapted to Reinformer-based architectures. This confirms that the benefits of language model initialization and prompt regularization extend beyond the standard Decision Transformer. We also include these results into our revised Table 1.
>
> |Task|Prompt-Rein|LPDT-Rein-Classifier|LPDT-Rein-InfoNCE|
> |-|-|-|-|
> |Cheetah-dir|991.10±30.59|1131.94±51.09|1113.96±20.52|
> |Cheetah-vel|-114.06±22.51|-95.02±10.58|-96.07±5.33|
> |Ant-dir|702.24±36.89|769.52±69.40|759.54±10.95|
> |Point-robot|-7.77±0.90|-7.30 ± 0.35|-7.37 ± 0.27|
> |MW Reach-v2|3008.25±77.81|3023.38 ± 178.49| 2614.41 ± 27.57|
>
> ---
> >**Q3**: Table 3 mainly shows the ablation results, so it is unnecessary to show Prompt-DT results. It's better to re-organize the table so that the comparison between w/ and w/o regularization is more obvious.
>
> **A3**: Thank you for the helpful suggestion. We agree that including Prompt-DT in Table 3 was unnecessary for the ablation study. In the revised manuscript, we have removed the Prompt-DT results from Tables 3 and 4 and reorganized the tables to make the comparison between the models with and without prompt regularization clearer.
>
> ---
> >**Q4**: As mentioned in the weakness, the authors should consider make the color and bold criterion the same across all tables. Currently in Table 1 the best two results are colored; in table 2 the best results are bold while in other tables no values are colored or bolded.
>
> **A4**: Thank you for the helpful suggestion. We have revised all tables to ensure consistent formatting of highlighted results. We use the green background to demonstrate the best scores and use the blue background for the second-best scores in the revised Tables 1,2,6 and 7. For the remaining tables, they are not colored as they intend to demonstrate the improvements compared to the ablation variants.
>
> ---
> >**Q5**: The authors report the values for different variants (Classifier/InfoNCE). It would be better if the authors could provide additional analysis on the performance of these two variants.
>
> **A5**: Thank you for the helpful suggestion. For the two prompt regularization variants (Classifier and InfoNCE), we present their performance in Table 3 of the manuscript. Overall, both variants show improvements over the versions without regularization across most environments, except for the Point-robot task. In both the LPDT and LPDT-Rein settings, the Classifier variant slightly outperforms InfoNCE. However, the Classifier variant requires task labels during training and cannot easily adapt to scenarios where new tasks are continuously collected without modifying the model structure. In contrast, the InfoNCE variant only requires grouping information (i.e., whether two samples are from the same task) and can therefore be more flexibly applied in such settings. The InfoNCE variant also demonstrates consistent improvements compared with the versions without regularization. We have included this analysis in Section 4.5 of the revised manuscript.

---

> > ### Comment · Reviewer_6SHM · 2025-10-23
> >
> > Thanks the authors for addressing my concerns. I do not have further questions.

---

### Review · Reviewer_vxMK · 2025-09-17

**Summary Of Contributions:**

This paper proposes the Language model-initialized Prompt Decision Transformer (LPDT), a novel framework for improving few-shot generalization in offline reinforcement learning (RL). The method leverages pretrained language models to provide strong prior knowledge, integrates Low-Rank Adaptation (LoRA) for efficient fine-tuning, and introduces prompt regularization to better distinguish between tasks. The approach is evaluated on MuJoCo and Meta-World ML1 tasks, and is further extended to Reinformer, showing strong performance with less training data compared to baselines. Extensive ablation studies validate the contributions of each component.

**Additional Comments:**

NA

**Audience:**

Yes

**Audience Explanation:**

A significant portion of the audience is likely to be interested in offline RL and in how large language models (LLMs) can be leveraged to improve the performance of offline RL algorithms.

**Claims And Evidence:**

Yes

**Claims Explanation:**

Extensive experiments verify the advantage of the proposed framework.

**Requested Changes:**

The writing and presentation of the paper are clear and well-structured. I mainly have the following concerns:

The method integrates LLM initialization, LoRA, and prompt regularization. To better evaluate the practical trade-offs of the proposed approach, it would be better to report the associated computational costs relative to existing baselines.

The paper claims that LPDT achieves performance comparable to Prompt-DT with only 10% of the data. However, this outcome appears only in limited cases, which makes the claim potentially overstated. Moreover, in Section 4.4, the reported results in the MW Reach-v2 task suggest that performance sometimes improves as the amount of training data decreases—a counterintuitive trend that warrants further discussion.

---

> ### Author Response · Authors · 2025-10-16
>
> We thank the reviewer for their valuable time and effort in providing detailed feedback on our work. We have uploaded a revised version with changes highlighted in blue text. We added the experiments on Prompt-Rein and reorganized tables for better readability. We also revised several unclear statements in Section 3.1, Section 3.3 and Section 4.5. In the following, we provide a detailed response to your questions and hope you find them satisfying.
>
> ---
> >**Q1**: The method integrates LLM initialization, LoRA, and prompt regularization. To better evaluate the practical trade-offs of the proposed approach, it would be better to report the associated computational costs relative to existing baselines.
>
> **A1**: Thank you for the valuable suggestion. To better evaluate the practical trade-offs introduced by our LPDT framework, we provided a detailed comparison of its computational costs relative to Prompt-DT and Meta-DT. We conducted experiments to measure the training and inference time and the number of trainable parameters for each method. We include these results in the following table. All experiments are conducted on a single NVIDIA A6000 GPU with Intel Xeon Ice Lake Gold 5317 processors. For the training time, we compare the methods by achieving the same performance on unseen test tasks. Since these methods have different performance on the Ant-dir, for a fair comparison, we choose the performance to achieve scores around 650, which is the performance of the prompt DT.
> From the table, our proposed LPDT can achieve a similar performance (around 650) compared to Prompt-DT with only a little more computation memory and time cost. Compared to the Meta-DT, we conclude that our LPDT variants achieve faster inference compared to the Meta-DT because we avoid the additional overhead on prompt selection as stated in the paper. It is also worth noting that Meta-DT needs to train a world model which has a longer training time. In addition to this, the LPDT-rein achieves around 2 times inference time compared to the LPDT. This is reasonable as the LPDT-rein needs to predict the return and action during the inference.
>
> | **Method**      |  **Transformer Trainable Parameters (M)**  |**Training Time** | **Inference Time** |
> | -------------- |  ---------------------------- |  ------------------------------ | ---------------------- |
> | **Prompt-DT**       |        0.595200        |          ~3.8h                      |         4.96s                      |
> | **Meta-DT**         |        0.793444                    |      ~14.7h                         |    10.41s                       |
> | **LPDT-Classifier** |   0.912384                        |       ~5.6h                         |         7.33s                 |
> | **LPDT-InfoNCE** |        0.912384                     |          ~5.7h                      |             7.82s                  |
> | **LPDT-Rein-Classifier** |         0.912384                     |      ~4.7h                          |   16.96s        |
> | **LPDT-Rein-InfoNCE** |         0.912384                     |             ~4.7h                   |           18.02s          |
>
> ---
> >**Q2**: The paper claims that LPDT achieves performance comparable to Prompt-DT with only 10% of the data. However, this outcome appears only in limited cases, which makes the claim potentially overstated. Moreover, in Section 4.4, the reported results in the MW Reach-v2 task suggest that performance sometimes improves as the amount of training data decreases—a counterintuitive trend that warrants further discussion.
>
> **A2**: We appreciate this comment and refer to Section 4.4 for a detailed discussion of this claim. Specifically, we demonstrate that our proposed LPDT can achieve performance comparable or superior to Prompt-DT on the Cheetah-vel and Ant-dir tasks while using only 10% of the data. We have also revised the claims in our abstract and introduction to be more detailed and avoid misunderstanding.
>
> Regarding the MW Reach-v2 task, we acknowledge that the performance exhibits significant fluctuations. We attribute this trend to the fact that all evaluated methods struggle on this particular task. For context, the appendix of Meta-DT [1] shows that an online SAC agent can quickly achieve returns of around 5000 with a few steps. In contrast, LPDT, Prompt-DT, and Meta-DT only reach approximately 3000, indicating that these approaches are less effective on this task. Therefore, we hypothesize that the observed fluctuations as the dataset decreases show the poor overall performance of all baseline methods on the MW Reach-v2 task.
>
> [1] Meta-DT: Offline Meta-RL as Conditional Sequence Modeling with World Model Disentanglement. 2024

---

> > ### Comment · Reviewer_vxMK · 2025-10-24
> >
> > Thank the authors for providing further experimental results. I have no further concerns.

---

### Review · Reviewer_dc7i · 2025-10-02

**Summary Of Contributions:**

This paper builds off of recent literature applying transformers to RL problems. Specifically, it aims to make Prompt-Decision Transformers better and more data efficient by initializing with pre-trained language models. Variants of this method are compared to pre-existing techniques, including under varying amounts of training data. Ablation studies are used to identify aspects of the method that are most important for performance.

**Audience:**

Yes

**Audience Explanation:**

Exploring how pre-trained LLMs may help prompt-based DTs is a reasonable thing to attempt, and so this paper would be of use to anyone interested in the outcome of that.

**Claims And Evidence:**

No

**Claims Explanation:**

I think there is a problem with the way results are analyzed and interpreted here. Specifically, the authors test 4 variants of their LPDT method. When they report whether LPDT beats other methods, they seem to consider LPDT a success if any one of these variants is better than existing methods. This is not a fair comparison; the other methods aren't allowed many variants, so they don't have as many chances to perform best. And indeed it is not the same LPDT variant that performs best across different tests. To make a claim about LPDT in total, it may be more appropriate to report the mean performance across all LPDT variants.

**Requested Changes:**

I don't think the manuscript mentions how the encoder and action weights are trained. Please include that.

It seems like the contrastive learning for the prompt encoder still requires task labels (to know if two samples are from same or different tasks), yet it is described as being relevant for when tasks aren't labeled. Please reconcile this.


The manuscript makes reference to LPDT possibly being more computationally efficient than other methods (for example, when contextualizing Meta-DTs performance by saying "It is worth noting that Meta-DT introduces an additional prompt selection stage at test time to select high-quality prompts, while our methods avoid this computation overhead."). Can the authors include the relative sizes of the models to understand computational costs?


It's not clear to me that these results represent significant differences, so this claim should be toned down: "showing that prompt regularization improves the model’s ability to distinguish between tasks and enhances overall performance." In general the authors also need to address how results around the four different variants of LPDT are being handled, as mentioned above.

---

> ### Author Response · Authors · 2025-10-16
>
> We thank the reviewer for their valuable time and effort in providing detailed feedback on our work. We have uploaded a revised version with changes highlighted in blue text. We added the experiments on Prompt-Rein and reorganized tables for better readability. We also revised several unclear statements in Section 3.1, Section 3.3 and Section 4.5. In the following, we provide a detailed response to your questions and hope you find them satisfying.
>
> ---
> >**Q1:** To make a claim about LPDT in total, it may be more appropriate to report the mean performance across all LPDT variants.
>
> **A1:** Thank you for your advice. We calculate the mean of our proposed variants in the revision paper Table 1 and Table 2. We also present Table 2 as follows for your reference, which shows the comparison of our proposed LPDT with the baseline methods under different ratios of the dataset. We illustrate the performance of our algorithms under different ratios from 1.0 to 0.01. According to the following tables, we can conclude that our proposed LPDT with the mean of all variants can also outperform Meta-DT under the {0.5,0.1,0.05}. This demonstrates the effectiveness of our methods.
>
> Furthermore, we would also like to make a clarification on why we present different variants of LPDT. We provide four variants of our LPDT to show the flexibility of our proposed framework. We demonstrate the performance of four variants to show the effectiveness of our methods. Current performance of LPDT can be further improved with more advanced prompt decision transformer variants and prompt regularization methods.
> |Dataset Ratio|Algorithm|Cheetah-dir|Cheetah-vel|Ant-dir|Point-robot|MW Reach-v2|
> |-|-|-|-|-|-|-|
> |1.0|Prompt-DT|960.32±17.07|-133.78±18.24|678.07±68.74|-7.99±0.46|2906.67±33.16|
> |1.0|Meta-DT|874.91±73.45|-52.42±8.11|961.27±18.07|-6.90±0.11|1418.39±6.01|
> |1.0|LPDT-Classifier|1121.68±25.13|-55.15±15.83|782.25±30.39|-8.19±0.46|2903.43±55.09|
> |1.0|LPDT-InfoNCE|1118.95±35.14|-57.56±9.55|775.79±35.28|-9.06±0.22|2946.88±92.18|
> |1.0|LPDT-Rein-Classifier|1131.94±51.09|-95.02±10.58|769.52±69.40|-7.30±0.35|3023.38±178.49|
> |1.0|LPDT-Rein-InfoNCE|1113.96±20.52|-96.07±5.33|759.54±10.95|-7.37±0.27|2614.41±27.57|
> |1.0|Mean of Variants|1121.63|-75.95|771.78|-7.98|2872.03|
> |0.5|Prompt-DT|691.05±72.87|-69.93±7.59|663.49±17.06|-6.53±0.41|3062.35±15.89|
> |0.5|Meta-DT|202.82±33.91|-75.61±7.77|508.73±61.73|-14.07±0.26|2085.79±2.62|
> |0.5|LPDT-Classifier|1074.20±27.26|-55.19±10.94|705.99±51.27|-6.04±0.56|2426.50±125.30|
> |0.5|LPDT-InfoNCE|1102.18±43.18|-64.23±5.76|714.31±67.15|-6.97±1.55|2439.34±131.92|
> |0.5|LPDT-Rein-Classifier|949.85±25.68|-101.84±4.23|746.45±56.32|-7.83±0.94|2841.04±136.94|
> |0.5|LPDT-Rein-InfoNCE|899.25±32.85|-106.94±8.52|750.38±70.14|-7.47±0.44|2902.60±92.70|
> |0.5|Mean of Variants|1006.37|-82.05|729.28|-7.08|2652.37|
> |0.1|Prompt-DT|193.69±9.06|-72.47±3.14|598.29±21.61|-8.94±0.43|3541.67±75.44|
> |0.1|Meta-DT|77.42±11.90|-114.32±10.24|509.25±51.03|-16.68±1.64|2040.93±166.34|
> |0.1|LPDT-Classifier|750.90±88.86|-65.23±19.16|688.21±36.78|-8.46±0.34|2519.92±152.28|
> |0.1|LPDT-InfoNCE|666.13±56.85|-77.08±17.71|660.63±52.00|-8.19±0.91|2577.42±60.06|
> |0.1|LPDT-Rein-Classifier|622.95±45.98|-107.29±15.12|687.71±68.32|-9.24±0.54|2781.77±170.80|
> |0.1|LPDT-Rein-InfoNCE|614.52±85.21|-112.69±14.25|641.87±45.96|-9.61±1.23|2827.77±154.02|
> |0.1|Mean of Variants|663.63|-90.57|669.61|-8.88|2676.72|
> |0.05|Prompt-DT|112.40±14.43|-97.29±9.90|491.32±19.94|-8.26±0.07|3320.88±77.59|
> |0.05|Meta-DT|37.87±19.15|-118.15±13.35|309.68±63.76|-18.27±1.67|2004.85±44.59|
> |0.05|LPDT-Classifier|386.01±84.45|-103.43±5.45|488.54±43.80|-7.80±1.22|2510.56±120.62|
> |0.05|LPDT-InfoNCE|320.28±79.24|-102.50±5.09|456.47±65.82|-8.32±0.45|2592.62±153.21|
> |0.05|LPDT-Rein-Classifier|252.33±69.15|-113.70±6.33|534.28±52.31|-10.22±1.23|2508.67±124.36|
> |0.05|LPDT-Rein-InfoNCE|279.48±45.58|-112.68±5.48|458.98±43.64|-10.50±1.44|2295.21±91.08|
> |0.05|Mean of Variants|309.53|-108.08|484.57|-9.21|2476.77|
> |0.01|Prompt-DT|18.97±4.52|-99.89±2.85|183.55±11.39|-8.91±0.41|2551.17±122.52|
> |0.01|Meta-DT|30.88±21.07|-173.93±28.09|90.86±0.91|-17.28±2.01|1934.47±28.61|
> |0.01|LPDT-Classifier|17.80±12.73|-121.84±4.12|175.82±18.61|-10.56±0.21|2132.60±165.87|
> |0.01|LPDT-InfoNCE|10.54±6.75|-127.81±3.26|130.76±3.64|-11.52±1.80|2390.82±79.37|
> |0.01|LPDT-Rein-Classifier|25.42±14.50|-125.22±2.58|264.61±12.04|-11.15±0.47|2508.67±124.36|
> |0.01|LPDT-Rein-InfoNCE|9.26±10.32|-133.44±4.71|270.43±8.36|-10.93±1.12|2449.91±91.08|
> |0.01|Mean of Variants|15.76|-127.08|210.41|-11.04|2370.50|

---

> ### Author Response · Authors · 2025-10-16
>
> ---
> >**Q2**: I don't think the manuscript mentions how the encoder and action weights are trained. Please include that.
>
>
> **A2:** We have discussed this in Section 3.1 on page 5 in the previous manuscript. We replace the word embedding layer and output head with the newly initialized input projection layer and output head. To make it clearer, we further added the following sentences in our revision:
>
> “Since the original token embedding and output head are trained on the text corpus, they can not be adapted to the RL trajectories. We need to train our own input projection layer $E_{RL}$ and output head $W_{act}$ from scratch on the RL trajectories dataset. To be more specific, the input projection layer $E_{RL}$ and output head $W_{act}$ are randomly initialized with the frozen pretrained language model backbone. Then the input projection and output head are fully adapted to the RL task trajectories and output the action for the RL problems.”
>
> In addition to this, we also add the following revision in Section 3.3:
>
> “The encoder $g_{\phi}$ is randomly initialized and jointly trained by Equation (3.5) from scratch.”
>
> To summarize, in our proposed methods, the input projection layer, prompt encoder, and output head are randomly initialized and trained from scratch with the frozen pretrained backbone and LoRA trainable parameters.
>
> ---
> >**Q3**: It seems like the contrastive learning for the prompt encoder still requires task labels (to know if two samples are from same or different tasks), yet it is described as being relevant for when tasks aren't labeled. Please reconcile this.
>
> A3: We would like to clarify that the contrastive learning for the prompt encoder only requires distinguishing between positive and negative pairs, i.e., identifying whether two prompts originate from the same or different tasks, but it **does not** rely on explicit task labels. In practice, we only need to know whether two samples come from the same task, without identifying their exact task labels. This is consistent with the literature of contrastive learning. Unlike supervised classifier regularization, the contrastive objective relies on the positive or negative information rather than explicit task labels for each task during optimization. This is achieved by assuming that all data points collected within the same batch originate from the same underlying task. Furthermore, this design also allows the method to be adapted to settings where tasks are continuously collected during training without changing the model structure. We have clarified this point in the revised Section 3.3 to avoid any possible misunderstanding.

---

> ### Author Response · Authors · 2025-10-16
>
> ---
> >**Q4**: The manuscript makes reference to LPDT possibly being more computationally efficient than other methods (for example, when contextualizing Meta-DTs performance by saying "It is worth noting that Meta-DT introduces an additional prompt selection stage at test time to select high-quality prompts, while our methods avoid this computation overhead."). Can the authors include the relative sizes of the models to understand computational costs?
>
>
>
> **A4:** Thank you for your question. We would clarify that we only claim that our proposed methods can avoid the computational overhead caused by the prompt selection stage of Meta-DT. To support our claim, we conducted additional experiments to illustrate the computational cost of our four variants compared to the Prompt-DT and Meta-DT. We demonstrate the trainable parameters of the transformer, the training time to achieve the same performance, and the inference time for one trajectory in each test task. All experiments are conducted on a single NVIDIA A6000 GPU with Intel Xeon Ice Lake Gold 5317 processors. For the training time, we compare the methods by achieving the same performance on unseen test tasks. Since these methods have different performance on the Ant-dir, for a fair comparison, we choose the performance to achieve scores around 650, which is the performance of the prompt DT.
>
> Our LPDT employs parameter-efficient fine-tuning via LoRA, which substantially reduces the number of trainable parameters compared to full fine-tuning, and it also avoids the additional prompt-selection stage required by Meta-DT during inference. The results of the training under the Ant-dir are summarized in the following table. From the table, we conclude that our LPDT variants achieve faster inference compared to the Meta-DT because we avoid the additional overhead on prompt selection. The LPDT-rein achieves around two times inference time compared to the LPDT is reasonable as the LPDT-rein needs to predict the return and action during the inference. In addition to this, our proposed LPDT achieves a similar performance compared to Prompt-DT with only a little more computation memory and time cost. It is also worth noting that Meta-DT needs to train a world model which has a longer training time.
>
> | **Method**      |  **Transformer Trainable Parameters (M)**  |**Training Time** | **Inference Time** |
> | -------------- |  ---------------------------- |  ------------------------------ | ---------------------- |
> | **Prompt-DT**       |        0.595200        |          ~3.8h                      |         4.96s                      |
> | **Meta-DT**         |        0.793444                    |      ~14.7h                         |    10.41s                       |
> | **LPDT-Classifier** |   0.912384                        |       ~5.6h                         |         7.33s                 |
> | **LPDT-InfoNCE** |        0.912384                     |          ~5.7h                      |             7.82s                  |
> | **LPDT-Rein-Classifier** |         0.912384                     |      ~4.7h                          |   16.96s        |
> | **LPDT-Rein-InfoNCE** |         0.912384                     |             ~4.7h                   |           18.02s          |

---

> ### Author Response · Authors · 2025-10-16
>
> ---
> >**Q5**: It's not clear to me that these results represent significant differences, so this claim should be toned down: "showing that prompt regularization improves the model’s ability to distinguish between tasks and enhances overall performance." In general, the authors also need to address how results around the four different variants of LPDT are being handled, as mentioned above.
>
> **A5**: We appreciate the reviewer’s helpful comment. We reformulated Table 3 in our revision and made it clearer. In revision, we remove Prompt-DT and add new columns of the mean for LPDT and LPDT-Rein variants respectively. We also include the revised table as follows. From the mean performance, we can also find that our proposed prompt regularization methods have improved performance compared to the variant without the prompt regularization except for Point-robot. We have revised our claim in our revision to “These results show that prompt regularization improves the model's ability to distinguish between tasks and demonstrates improvements in most settings.”
>
> | Task         | LPDT w/o regularization | LPDT-Classifier | LPDT-InfoNCE | Mean | LPDT-Rein w/o regularization | LPDT-Rein-Classifier | LPDT-Rein-InfoNCE | Mean |
> |---------------|--------------------|----------------------|------------------|------|--------------------|---------------------------|------------------------|------|
> | **Cheetah-dir** | 1109.54 ± 20.04 | 1121.68 ± 25.13 | 1118.95 ± 35.14 | **1120.32** | 1123.51 ± 9.67 | 1131.94 ± 51.09 | 1113.96 ± 20.52 | **1122.95** |
> | **Cheetah-vel** | -56.38 ± 7.53 | -55.15 ± 15.83 | -57.56 ± 9.55 | **-56.40** | -95.39 ± 23.80 | -95.02 ± 10.58 | -96.07 ± 5.33 | **-95.55** |
> | **Ant-dir** | 762.19 ± 59.22 | 782.25 ± 30.39 | 775.79 ± 35.28 | **779.02** | 718.78 ± 43.41 | 769.52 ± 69.40 | 759.54 ± 10.95 | **764.53** |
> | **Point-robot** | -6.37 ± 0.44 | -8.19 ± 0.46 | -9.06 ± 0.22 | **-8.63** | -6.12 ± 0.26 | -7.30 ± 0.35 | -7.37 ± 0.27 | **-7.34** |
> | **MW reach-v2** | 2808.15 ± 189.82 | 2903.43 ± 55.09 | 2946.88 ± 92.18 | **2925.16** | 2252.68 ± 156.39 | 3023.38 ± 178.49 | 2614.41 ± 27.57 | **2818.90** |

---

> > ### Comment · Reviewer_dc7i · 2025-10-23
> >
> > I thank the authors for their work in addressing my comments; I think the paper is improved as a result.

---

### Decision · Action_Editor_i91t · 2025-10-30

**Recommendation:** Accept with minor revision

**Additional Comments:**

The authors are commended for their responsive revisions and for conducting additional experiments that strengthened the paper, particularly the inclusion of mean performance metrics and the computational cost analysis. The work presents a promising new direction for integrating large-scale pre-training with Decision Transformers in the few-shot setting.

**Audience:**

Yes

**Audience Explanation:**

The findings of this paper will be of interest to the TMLR audience, especially those working in offline RL, meta-RL, and the intersection of LLMs with RL/Decision Transformers. The work introduces a novel approach to incorporating pre-trained language models into the Prompt Decision Transformer framework, offering valuable insights and practical methods to improve data efficiency and few-shot generalization, two critical challenges in the field.

**Claims And Evidence:**

Yes

**Claims Explanation:**

The claims regarding the proposed Language model-initialized Prompt Decision Transformer (LPDT) framework are well-supported by convincing empirical studies. The authors conducted experiments and ablation studies, which clearly validate the effectiveness of the key components: PLM initialization, LoRA for efficient fine-tuning, and prompt regularization. Specifically, the experiments demonstrate that LPDT variants can achieve performance comparable to or superior to baselines like Prompt-DT and Meta-DT, particularly in few-shot settings and with less training data. Concerns raised by reviewers regarding data analysis (e.g., aggregating LPDT variants) and computational costs have been addressed in the revision.

---

> ### Author Response · Authors · 2025-11-12
>
> Thank you for your valuable feedback. We have carefully revised the manuscript in response to the comments from both the reviewers and the editor and have submitted the camera-ready version.
>
> In particular, we added more details in Section 3.1 to clarify the training of the input projection layer and the output head, and we also expanded Section 3.3 on contrastive learning based prompt regularization to avoid potential misunderstandings.
>
> In Section 4, we included the mean metrics of our LPDT variants in Tables 1, 2, and 3 to make the comparison with the baselines clearer.
>
> Additionally, in Section 4.5, we added a computation analysis to show the training cost and inference time compared to Prompt-DT and Meta-DT.